# Changes in Human Erythrocyte Membrane Exposed to Aqueous and Ethanolic Extracts from *Uncaria tomentosa*

**DOI:** 10.3390/molecules26113189

**Published:** 2021-05-26

**Authors:** Piotr Duchnowicz, Radosław Pilarski, Jaromir Michałowicz, Bożena Bukowska

**Affiliations:** 1Department of Biophysics of Environmental Pollution, Faculty of Biology and Environmental Protection, University of Lodz, 141/143 Pomorska St., 90-236 Lodz, Poland; jaromir.michalowicz@biol.uni.lodz.pl (J.M.); bozena.bukowska@biol.uni.lodz.pl (B.B.); 2Polish Academy of Sciences, Institute of Bioorganic Chemistry, Noskowskiego 12/14 Str., 61-704 Poznań, Poland; rapil@ibch.poznan.pl

**Keywords:** erythrocytes, membrane fluidity, osmotic resistance, *Uncaria tomentosa*, extracts

## Abstract

*Uncaria tomentosa* (Willd.) DC is a woody climber species originating from South and Central America that has been used in the therapy of asthma, rheumatism, hypertension, and blood purification. Our previous study showed that *U. tomentosa* extracts altered human erythrocyte shape, which could be due to incorporation of the compounds contained in extracts into the erythrocyte membrane. The aim of the present study was to determine how the compounds contained in *U. tomentosa* extracts incorporate into the human erythrocyte membrane. The study has assessed the effect of aqueous and ethanolic extracts from leaves and bark of *U. tomentosa* on the osmotic resistance of the human erythrocyte, the viscosity of erythrocyte interior, and the fluidity of erythrocyte plasma membrane. Human erythrocytes were incubated with the studied extracts in the concentrations of 100, 250, and 500 µg/mL for 2, 5, and 24 h. All extracts tested caused a decrease in erythrocyte membrane fluidity and increased erythrocyte osmotic sensitivity. The ethanolic extracts from the bark and leaves increased viscosity of the erythrocytes. The largest changes in the studied parameters were observed in the cells incubated with bark ethanolic extract. We consider that the compounds from *U. tomentosa* extracts mainly build into the outer, hydrophilic monolayer of the erythrocyte membrane, thus protecting the erythrocytes against the adverse effects of oxidative stress.

## 1. Introduction

*Uncaria tomentosa* extracts are commonly used to support the treatment of various diseases due to their antioxidant [1,2,3,4] and immunostimulating [5,6] activities. Extracts from *U. tomentosa* have also been shown to exhibit anticancer [7,8,9], antimutagenic [10,11] and antiherpetic activities [11].

Ethnomedicinal surveys and other studies have shown numerous applications of this plant in traditional medicine of different cultures, e.g., the treatment of inflammation, arthritis, asthma, cancer, gastric ulcers, urinary tract diseases and hemorrhages [12,13]. Such diverse properties are due to the complex chemical composition of *U. tomentosa* being mainly associated with high content of various biologically active secondary metabolites, mostly phenolic and alkaloids [14].

Many papers have also indicated the beneficial action (without any side effects) of *U. tomentosa* extracts (used in therapeutic dosages) in blood cells in both in vivo [15,16] and in vitro studies [17,18,19]. Sheng et al. [20] did not observe any changes in blood presentation following administration of high concentrations of *U. tomentosa* extract C-MED-100TM to healthy volunteers. They did not observe any statistically significant changes, including hemoglobin level, hematocrit value, and mean red blood cell (RBC) volume, which rather excluded possible premature elimination of RBCs exposed to *U. tomentosa* extracts. Similarly, clinical trials conducted by Piscoya et al. [21] on patients with knee osteoarthritis, regarding the evaluation of the efficacy of *Uncaria gujanensis* in the therapy of this disease, showed no changes in hematocrit, hemoglobin level, as well as the erythrocyte sedimentation rate (ESR) value, compared to a group of patients receiving placebo. Nevertheless, further studies on larger human cohorts are necessary to exclude the possibility of elimination of premature erythrocytes from circulation of humans exposed to *Uncaria* species. Our earlier studies have shown that ethanolic and aqueous extracts from the leaves and bark of *U. tomentosa* even at a very high concentration of 500 µg/mL were not toxic to the erythrocytes. We have also observed that the studied extracts alleviated hemoglobin oxidation and lipid peroxidation as well as decreased ROS formation and hemolysis induced by a toxicant 2,4-dichlorophenol (2,4-DCP) [17]. Additionally, we found a protective effect of *U. tomentosa* extracts on the activity of catalase in human erythrocytes incubated with 2,4-DCP and catechol [22].

The physiological concentration of plant polyphenols is difficult to assess. In the literature, usually the information about recommended doses of preparations or the concentrations of tested compounds (for humans 210–500 mg/day or for animals 350–500 mg/kg body mass) is given [23,24].

Some of the compounds are absorbed without changing their form, while others, especially in the form of glucosides, during absorption are depriving of all or a part of sugar residues. The intestinal flora also participates in the transformation of plant polyphenols.

Most of the absorbed dose is excreted from the body via urine and faces. It is estimated that no more than 10–20% of the ingested dose of quercetin and isoflavones (genistein and daidzein) is detected in plasma after 1–3 h of administration. Anthocyanins are considered to be less efficiently absorbed–their absorption is estimated at 1–2% of the initial dose [25].

Oxidative stress occurs when the level of ROS in a cell overwhelms the cell’s antioxidant defense system. Red blood cell is particularly susceptible to oxidative stress due to the high content of polyunsaturated fatty acids present in its membrane and the auto-oxidation of hemoglobin. Human erythrocyte membrane is composed from saturated fatty acids, monounsaturated fatty acids, polyunsaturated fatty acids and long-chain N-3 polyunsaturated fatty acids [26]. Oxidizing agents change membrane stiffness, alter membrane dipole potential and cause the formation of spectrin–hemoglobin complexes [27]. To cope with oxidative stress, the erythrocytes are equipped with a specialized cytoskeleton that provides their mechanical stability and flexibility necessary to withstand forces occurring during circulation. Erythrocytes membrane is composed of a lipid bilayer, transmembrane proteins and a filamentous meshwork of proteins (such as spectrin, F-actin, protein 4.1, and actin-binding proteins dematin, adducin, tropomyosin, and tropomodulin) that form a membrane skeleton along the entire cytoplasmic surface of their membrane [28]. ROS are also capable of inducing eryptosis by changes in membrane phosphatidylserine (PS) translocation [29]. Oxidative stress may also lead to exhaustion of cellular reductive components and cause structural and functional changes in red blood cell membrane: degradation of phosphatidylserine and sialic acid, which are the main components establishing the negative charge on the outer side of the membrane [30].

It has been proven that polyphenols isolated from *U. tomentosa* exhibit very strong antioxidative properties. Several reports have shown that aqueous and methanolic extracts obtained from different parts of *U. tomentosa* protected cells from free radicals production and lipid peroxidation in vitro, as well as inhibited free radical-induced DNA damage and cell death [2,3,31]. In our previous study we also showed that *U. tomentosa* extracts under physiological conditions protected erythrocytes against ROS formation and hemolysis. Moreover, we demonstrated that ethanolic extracts from *U. tomentosa* exhibited the strongest protective effects [17].

To find out whether the tested compounds can penetrate the membrane (and at what depth they are located), the erythrocyte membrane fluidity was assessed. Many physical and chemical factors affect membrane fluidity [32,33]. Chemical agents include xenobiotics that cause lipid and protein oxidation and the compounds that intercalate into the membrane [34,35]. The incorporation of foreign compounds into the membrane structure can disrupt the arrangement of its elements, which usually results in a change in membrane fluidity. Changes in membrane fluidity may lead to erythrocyte dysfunction as well as to alterations in the rheological properties of erythrocyte [36]. On the other hand, the substances with antioxidant properties having the ability to penetrate membrane may exhibit a protective role towards the cell [37]. Therefore, to clarify whether the compounds contained in *U. tomentosa* extracts can build into the inner or/and outer membrane leaflet, we assessed alterations in membrane fluidity at its different levels as well as evaluated osmotic resistance and internal viscosity of erythrocyte. We chose relatively high concentrations (but not toxic as shown in previous studies) of the tested extracts to observe potential changes in the studied parameters as well as to determine location of the extract components in erythrocyte membrane.

## 2. Results

### 2.1. Membrane Fluidity

#### 2.1.1. Order Parameter S

After 2 h and 24 h of incubation of the erythrocytes with aqueous and ethanolic extracts from bark and leaves, statistically significant changes were noted in the order parameter S. Figure 1 shows the average and standard deviation values for four (2 h incubation) and five (24 h incubation) independent experiments. After 2 h of incubation, all extracts studied at 500 μg/mL caused an increase in the ordering parameter S in human RBCs. After 24 h of incubation, statistically significant changes the studied parameter were noted both at 250 μg/mL and 500 μg/mL of all examined extracts.

#### 2.1.2. Correlation Time τ_B_ and Time τ_C_

After 2 h of incubation, statistically significant changes were observed for the correlation time τ_B_ and time τ_C_ for the erythrocytes treated only with ethanolic extract at 500 μg/mL. After 24 h of incubation, no statistically significant changes in time τ_B_ and time τ_C_ were observed in RBCs treated with any tested extract (Table 1).

### 2.2. Internal Viscosity of Red Blood Cells

After 2 h and 24 h of incubation of RBCs with water and ethanolic extracts from the bark and the leaves in the concentrations range from 100 to 500 μg/mL, no statistically significant changes were observed in the relative viscosity of the erythrocyte interior (Figure 2). The graphs show the average and standard deviation values for four (2 h) and five (24 h) independent experiments.

It must be noted that after 2 h, and particularly after 24 h of incubation of the erythrocytes with ethanolic extracts from the bark, a tendency towards increased viscosity of the erythrocyte internum could be observed (the results were not statistically significant) (Figure 2).

### 2.3. Osmotic Fragility

Incubation of the erythrocytes with aqueous and ethanolic extracts from the bark and the leaves at a concentration of 100 μg/L was carried out for 2 h, 5 h, and 24 h. The osmotic resistance diagram (Figure 3) presents the average results for 5 independent experiments for each incubation time. For the readability of the chart, no standard deviations were plotted. The IC_50_ plots (Figure 4) show the calculated average and standard deviation values for 5 independent experiments for each incubation time.

After 2 h of incubation of the erythrocytes with the extracts tested, no changes in the osmotic sensitivity of these cells, and thus changes in IC_50_ parameter values were observed (Figure 3 and Figure 4). It was also noted that the tested extracts after 5 h of incubation increased the osmotic sensitivity of human erythrocytes. Except for the aqueous leaf extract, the other extracts sensitized RBCs to damage by osmotic stress. The erythrocytes incubated with the studied extracts underwent hemolysis at higher NaCl concentrations when compared to the control RBCs (Figure 3). An increase in the IC_50_ parameter values was also statistically significant for the erythrocytes incubated for 5 h with aqueous and ethanolic bark extracts and ethanolic leaf extract (Figure 4). After 24 h of incubation, greater sensitivity to osmotic stress was noted in the erythrocytes treated with all tested extracts in comparison to the control RBCs (Figure 3). Similarly, a statistically significant increase in IC_50_ values was noted in the erythrocytes incubated with all studied extracts from *U. tomentosa* (Figure 4).

## 3. Discussion

Earlier study showed that extracts of *U. tomentosa* changed morphology of human erythrocytes [19]. The formation of echinocytes was found, which indicated the incorporation of extracts’ components into the outer monolayer of the erythrocytes membrane [38,39]. Another study showed that the compounds contained in *U. tomentosa* extracts inhibited hemoglobin leakage (hemolysis) probably by sealing of the cell membrane [17]. One of the properties of the biological membrane, resulting from its structure and interaction between its components, is membrane fluidity. The fluidity of the membrane is inversely proportional to its viscosity.

Using paramagnetic resonance spectroscopy (EPR), the fluidity of erythrocyte membrane incubated with ethanol and aqueous extracts from bark and leaves was determined at two levels of the 5 and 16 carbon atoms of the fatty acid residue.

Based on the EPR spectra obtained for the 5-doxyl-stearic acid spin tag (5-DSA), the S-ordering parameter was calculated, which gave information on the fluidity changes at the level of 5 carbon atom of the fatty acid residue. For the ordering parameter S, a statistically significant increase in its value was observed after incubation of erythrocytes with ethanol and water extracts from the bark and leaves of *U. tomentosa*, which showed a decrease in membrane fluidity at the level of 5 carbon atom of a fatty acid residue. A statistically significant increase in the parameter S was observed after a 2-h incubation of erythrocytes with ethanolic extracts from the bark as well as water and ethanolic extract from leaves at 250 μg/mL and 500 μg/mL, while water extract from the bark only at 500 μg/mL caused changes in the discussing parameter (Figure 1A). Elongation of the incubation time up to 24 h caused a statistically significant increase in the S parameter in the erythrocytes incubated with all studied extracts starting from their concentration of 250 μg/mL (Figure 1B). The decrease in membrane fluidity at the level of 5 carbon atom of a fatty acid residue may have been linked to the effect of polyphenols on the structure of the plasma membrane as described by Duchnowicz et al. [40]. Research by Koren et al. [37] implied a beneficial effect of polyphenols on the total antioxidant capacity of the blood, and especially of the erythrocytes. They postulated that circulating erythrocytes and possibly other blood cells might be constantly coated by polyphenols from supplemented nutrients, which act as antioxidant depots, and thus can behave as protectors against the harmful consequences of oxidative stress. Polyphenolic compounds determined in *U. tomentosa* are classified as secondary metabolites that have a positive effect on the human body. They exhibit antioxidant activity, chelate metal ions (iron, copper) and support antioxidative enzymes activities. The protective effect of polyphenols on the human body is primarily associated with cell protection against oxidative stress, which may be associated with preventing the development of various diseases, including atherosclerosis or cancer [41,42]. Research carried out by Heitzman et al. [43] as well as our study [17] showed the presence of polyphenols in extracts of *U. tomentosa*. The authors, by analyzing various extracts, determined the highest content of polyphenols in the ethanol extract from the bark and the lowest in the aqueous extract from the leaves. Phenolic compounds present in *U. tomentosa* are mostly represented by tannins, flavonol derivatives, catechins, tannins (including catechins and epicatechins) and procyanidins. Polyphenols due to their structure have different water solubility, but they all dissolve well in organic solvents. Therefore, ethanol extracts from plants usually have a more diverse composition and a higher concentration of polyphenols than aqueous extracts.

In the case of the 16-doxyl stearic acid spin tag (16-DSA), two correlation times τ_B_ and τ_C_ were determined. After incubation of erythrocytes for 2 h with ethanol extract from the bark at a concentration of 500 μg/mL, an increase in the correlation time τ_B_ and time τ_C_ was observed. However, no statistically significant changes in these parameters were noted after 24 h of incubation. Other tested extracts did not cause statistically significant changes in τ_B_ correlation time both after 2 h and 24 h of incubation. An upward trend for this parameter was shown in erythrocytes incubated (particularly for 24 h) with all studied extracts at 500 µg/mL (Table 1). An increase in the correlation times τ_B_ and τ_C_ indicates a stiffening of the film at the level of 16 carbon atom of the fatty acid residue. However, no changes after 24 h of incubation suggests that this is a temporary effect caused by the components of the extract, which initially incorporate into the membrane, but after a longer incubation time, they are removed outside the cell. Such effects of polyphenols was described in the study of Tarahovsky et al. [44]. These authors suggested that flavonoids penetrated the hydrophobic site of the lipid bilayer, particularly into the compartments known as lipid rafts, or into hydrophobic protein pockets. Flavonoids after complexing with metals, still have the antioxidative potential, while their lipophilicity may increase, therefore facilitating the protection of membrane lipids against oxidation.

We suggest that polyphenols from the bark extract of *U. tomentosa* not only penetrated deeper levels of the erythrocyte membrane but were also transferred inside the cell, which was confirmed by the increase in RBC internal viscosity. One of the key parameters affecting the deformability of erythrocytes is its internal viscosity [45]. In this study, changes in erythrocyte viscosity were assessed by spin markers. There were no statistically significant changes in the relative viscosity of erythrocytes treated with water extracts from *U. tomentosa* both after 2-h and 24-h incubation, while ethanolic extracts induced changes in the discussing parameter (Figure 2). This may indicate that polyphenols contained in much higher concentrations in ethanolic extracts intensively penetrated RBC membrane causing changes in the viscosity of the interior of the erythrocyte. Erythrocytes incubated with water extracts could easily compensated changes in their viscosity, which were provoked by lower concentrations of polyphenolic substances (Figure 6).

It should be emphasized, however, that the location of polyphenols in the deeper layers of the cell membrane, and consequently, their penetration into the cell with simultaneous disturbance in RBC viscosity may occur only at a very high concentration of these substances that are not found in the human body even after therapeutic treatment with *U. tomentosa* extracts. At low concentrations, polyphenols are mainly located in the outer monolayer, as demonstrated by changes in the value of the parameter S. Alterations in the viscosity of the erythrocyte interior may be associated with a change in the shape of the erythrocytes. This was indicated by changes in the forward scatter channel (FSC) parameter obtained from flow cytometric analysis and photos achieved from a phase-contrast microscopic examination [19]. The formation of echinocytes may result from the placement of polyphenols present in *U. tomentosa* extracts both in the hydrophilic region and in the hydrophobic region of the outer cell membrane monolayer, as suggested by changes in the ordering parameter S and correlation times τ_B_ and τ_C_. Changes in red blood cell shape may be associated with water loss and the increase of the viscosity inside the cells. Additionally, they may be due to a decrease in the osmotic resistance of the erythrocytes. Placement of altered erythrocytes in an isotonic NaCl solution may lead to faster cell lysis (in comparison to control cells) because of increased effect of water and a limited possibility of deformation of the erythrocyte membrane, and therefore changes of the cell shape.

The results showed that depending on the incubation time, the extracts from *U. tomentosa* may have a different impact on the fluidity of the plasma membrane and osmotic resistance of erythrocytes.

Osmotic resistance of erythrocytes is a parameter that indicates the sensitivity of RBCs to the action of a hypotonic environment. By the exposure of the erythrocytes to the isotonic environment, a small osmotic movement of water in the cell membrane can be observed, which does not change the volume and shape of the cell. In the case of a hypotonic environment, water enters the cell, which causes its volume to increase, as well as changing the shape from a two-concave disk to a sphere-like shape. Cells can increase their volume only to a certain extent. Further increase in a volume results in a disruption of cell membrane integrity and release of cell content by osmotic lysis. After a 2-h of incubation of erythrocytes with extracts from *U. tomentosa*, no changes in osmotic sensitivity of RBCs (Figure 3 and Figure 4) as well as changes in the IC_50_ parameter were observed. However, after 24-h incubation, greater osmotic sensitivity of erythrocytes was observed, and thus an increase of IC_50_ value against the control was noted (Figure 3 and Figure 4). Stronger effects were observed after incubation of erythrocytes with ethanol extracts compared to aqueous extracts. This is probably due to the fact that ethanol extracts have more polyphenol compounds and tannins [17]. Similar changes (an increase in IC_50_ value) after longer incubation time (90 min at 37 °C) of erythrocytes with catechins were observed by Naparlo et al. [46]. These authors suggested that probably during prolonged incubation, catechins are taken up by erythrocytes, which may lead to an increase in the osmolarity of cell interior, and thus easier hemolysis in hypotonic solutions due to a greater difference in osmolarity between cell interior and exterior. The second explanation for such changes was suggestion that catechins generate ROS during prolonged incubation, e.g., hydrogen peroxide, which may damage erythrocyte membrane and increase its fragility, including osmotic fragility.

Location of the compounds at the level of 5 carbon of the erythrocyte fatty acid chain resulted in an increased osmotic sensitivity of these cells (shift of osmotic resistance curves towards higher NaCl concentration values, increase in IC_50_ values). However, in previous experiments conducted by Bors et al. [17] under isotonic conditions (0.9% NaCl), *U. tomentosa* extracts inhibited hemolysis induced by toxicants, such as 2,4-DCP and catechol. These researchers also demonstrated the protective role of *U. tomentosa* extracts against the oxidative action of 2,4-DCP in human erythrocytes manifested in reducing methemoglobin and ROS levels. Thus, the final effect of the extracts in the body is probably positive. In the in vivo system, extracts from *U. tomentosa* (at low concentrations) incorporate into the membrane (without interfering with its structure) and exhibit antioxidant protective properties [12,47]. The results of the studies on rats as well as on healthy volunteers who were administered *U. tomentosa* did not show any changes in red blood cell membrane or other changes in the erythrocytes, which excluded the possibility of premature removal of these cells from circulation [20,48].

We claim that binding of polyphenols to the erythrocyte surface may be responsible for their protective properties (observed in earlier studies). Polyphenols, which are the most often represented in extracts from *U. tomentosa*, can interact directly with biological membranes, and they also act as scavengers of free radicals and inhibitors of lipid peroxidation, increasing cell resistance to oxidative damage. This was confirmed by the latest research conducted on bioactive membranes. Kim et al. [49] added *U. tomentosa* extract in the process of nanostructured alginate membrane formulation to improve its biological activities. The authors showed that membrane containing component from *U. tomentosa* possessed an excellent antioxidant activity, showing almost 100% of the reduction capacity of free radicals in 15 min.

The limitation of this study is a lack of evaluation of the effect of tested extracts on the fatty acid composition in erythrocyte membrane. Human erythrocyte membrane is composed from saturated fatty acids (palmitic, stearic), monounsaturated fatty acids (palmitoleic, vaccenic, *cis*-oleic, *trans*-oleic), polyunsaturated fatty acids (*cis*-linoleic, *trans*-inoleic, linolenic, eicosadenoic, eicosatrienoic, arachosatrienoic) and long-chain N-3 polyunsaturated fatty acids (eicosapentaenoic, docosahexaenoic).

However, in this study, the extracts were applied to erythrocytes obtained from healthy donors. Thus, it can be assumed that the lipid composition of the erythrocyte membrane is typical for normal cells [26]. Since human mature erythrocytes do not have a nucleus, they do not synthesize cellular components, including fatty acids. For this reason, the lipid composition should not change as opposed to in vivo studies: in squamous cell carcinoma of the skin [26] or during supplementation [50,51]. The flip-flop process, lipid oxidation, change of interactions in the lipid domains and their distribution are possible, but as lipid biosynthesis in erythrocytes does not occur, the qualitative and quantitative composition of the erythrocyte membrane lipids under the influence of extracts tested added in vitro is very unlikely to occur.

In summary, the extracts of *U. tomentosa* mainly temporarily incorporate into the membrane, and exactly in its outer monolayer. However, at very high concentrations, they can penetrate deeper into the membrane and ultimately alter the viscosity of the erythrocytes. Based on the obtained results as well as literature data, we consider that physiological low concentrations of the compounds contained in extracts from *U. tomentosa* will be in the outer monolayer of cell membrane and will protect erythrocytes against oxidative stress and damage.

## 4. Materials and Methods

### 4.1. Chemicals

5-Doxylstearic acid (5-DSA), 16-doxylstearic acid (16-DSA), 4-hydroxy-2,2,6,6-tetramethylopiperidine-1-oxyl (TEMPOL) were bought from Sigma-Aldrich, Saint Louis, MI, USA. Other chemicals were obtained from Roth, Karlsruhe, Germany and Avantor Performance Materials Poland, Gliwice, Poland.

### 4.2. Biological Material

#### 4.2.1. Plant

The raw material (bark and leaves) of *U. tomentosa* originated from Instituto Peruano de Investigaciones Fitoterapia Andina in Lima, Peru, was kindly supplied by Wilcaccora Łomianki Centre, Poland. The general characteristics of this material were described previously by Pilarski et al. [3].

The voucher materials were deposited at the Laboratory of Phytochemistry, Institute of Bioorganic Chemistry, Polish Academy of Sciences, Poznań, Poland.

Preparation of samples: one gram of the raw material (bark and leaves) of *U. tomentosa* was extracted with a volume of 10 mL of water or 96% ethanol at 37 °C for 8 h. Then, the extracts were centrifuged at 4000 rpm for 15 min. Supernatants were evaporated with Speed-Vac to dry mass, and next desiccated with P_2_O_5_. At these conditions, about 111 mg of preparation per gram of bark and about 158 mg of preparation per gram of leaves in the case of aqueous extraction were covered. During extraction with 96% ethanol, 123 mg of preparation per gram of bark and 158 mg of preparation per gram of leaves were achieved. Four extracts were obtained, i.e., ethanolic extract from the bark (B_et_), aqueous extract from the bark (B_aq_), ethanolic extract from leaves (L_et_), and aqueous extract from leaves (L_aq_). Powdered preparations were stored in a refrigerator in tightly closed containers, from which stock solutions were prepared for each experiment [19].

The qualitative and quantitative composition of oxindole alkaloids contained in *U. tomentosa* extracts was determined by means of HPLC fingerprint analysis. The general characteristics of the quantitative alkaloids of this material were described previously by us in the publication of Bors et al. [18].

Uncarine F, speciophylline, mitraphylline, pteropodine/isomitraphylline, and isopteropodine were detected in *U. tomentosa* extracts. Mitraphylline and pteropodine/isomitraphylline are alkaloids, which are dominant in leaf extracts, while pteropodine/isomitraphylline are dominant in bark extracts. Ethanolic extracts were characterized by a higher alkaloid content than aqueous extracts (Figure 5).

The general characteristics of the quantitative determination of polyphenols in *U. tomentosa* extracts were described previously by us [17]. HPLC/DAD analysis revealed that bark ethanolic and aqueous extracts contained mainly Procyanidin B2 and C1 as well as (–)-epicatechin. Caffeic acid derivatives were only determined in aqueous extracts from leaves, whereas chlorogenic acid, flavonols, and polymeric proanthocyanidins were contained in all studied extracts. The highest total concentrations of free phenols were determined in the ethanolic extracts from the bark. High content of phenolic compounds was also detected in the ethanolic extract from leaves, whereas lower phenolic concentration was detected in aqueous extract from the bark and ethanolic extracts from leaves (Figure 6).

Finally, all extracts were dissolved in dimethyl sulfoxide (DMSO). The obtained solution was added in a small volume of 1 µL to 1 mL of erythrocytes suspension (DMSO concentration did not influence cell membrane). The erythrocytes supplemented only with DMSO were used as controls.

#### 4.2.2. Isolation of Erythrocytes

Leucocyte-buffy coat was obtained from blood collected in Blood Bank in Lodz, Poland. The leucocyte-buffy coat was centrifuged (3000 rpm for 10 min at 4 °C) and washed twice with phosphate-buffered saline (pH 7.4).

Isolated erythrocytes suspended in PBS were incubated for 2, 5, and 24 h with four types of extracts (ethanolic and aqueous extracts from the bark and ethanolic and aqueous extracts from the leaves) at the concentrations of 100 μg/mL, 250 μg/mL and 500 μg/mL. Previous studies have shown that the studied extracts at these concentrations did not cause damage to erythrocytes, while they exhibited protective effect on RBCs [17,19,22].

### 4.3. Membrane Fluidity

Fluidity of erythrocyte membrane was determined by means of electron paramagnetic resonance (EPR) spectroscopy (Brucker 300 Spectrometer, Germany) using 5-doxylstearic acid (5-DSA) and 16-doxylstearic acid (16-DSA) spin labels. From the EPR spectra obtained for the 5-DSA spin label, the ordering parameter S was calculated. The correlation times τ_B_ and τ_C_ were calculated from the EPR spectra obtained for the 16-DSA spin label. The ordering parameter S as well as the correlation times τ_B_ and τ_C_ show the inverse correlation with membrane fluidity. Order parameter S and the correlation times τ_B_ and τ_C_ were calculated as described in the study of Koter et al. [52].

### 4.4. Internal Viscosity of Red Blood Cells

The TEMPOL spin label was used to determine the internal viscosity of the erythrocytes [53]. TEMPOL solution (20 mM) dissolved in buffered saline (PBS) was added to the erythrocytes suspension with 50% hematocrit. After a 30-min incubation, the erythrocytes were washed three times with 80 mM K_3_Fe(CN)_6_ solution to reduce the signal from the spin label located in the intercellular space. Based on the spectrum obtained for TEMPOL in water and in erythrocytes, the rotational correlation time was determined according to the formula:(1)$$\tau =\left[{\left(\frac{h_{0}}{h_{-1}}\right)}^{-\frac{1}{2}}-1\right]$$
where: *k*—constant for the nitroxide spin label, *k* = 6.5 × 10^−9^ s mT^–1^, *h*_0_—height of the center line, *h*_−1_—height of high-field line, *W*_0_—centerline width.

The relative viscosity was determined from the formula:(2)$$\frac{\tau_{R}}{\tau_{B}}=\frac{\eta_{C}}{\eta_{B}}$$
where: *τ_R_*—rotational correlation time of the marker in erythrocytes, *τ_B_*—rotational correlation time of the marker in PBS, *η_C_*—internal viscosity of erythrocyte, *η_B_*—PBS viscosity (value 1 was assumed).

### 4.5. Osmotic Fragility

After incubation of the erythrocytes with extracts from *U. tomentosa*, the cells were centrifuged at 3000 rpm for 10 min at 4 °C. The cells were then washed with 0.9% NaCl solution. This step was repeated twice. After removing the supernatant, RBCs were suspended in a NaCl solution in the concentration range from 0.2 to 0.9% (15 points were selected). The cells were incubated for 30 min at 37 °C, and then centrifuged at 3000 rpm for 10 min at 4 °C. Optical density of the supernatant was measured at 540 nm (*A_probe_*) using a microplate absorbance reader (BioTek Elx808, BIOTEK).

Hemolysis in each tube was expressed as a percentage, taking hemolysis in distilled water as 100% (*A*_100%_):(3)$$\mathrm{Hemolysis}\;\mathrm{percentage}\;(\%)=\frac{A_{probe}}{A_{100\%}}\times 100$$

### 4.6. Statistical Analysis

The results were presented as mean ± SD achieved from 4 to 6 individual experiments (4–6 blood donors), whereas for each individual (donor), the experimental point was a mean value of 2–3 replications. Multiply comparisons between group mean differences were analyzed by one-way analysis of variance (ANOVA) followed by Tukey post hoc test. The differences were considered statistically significant for *p* < 0.05. All statistical analyses were performed using STATISTICA software (StatSoft, Inc, Tulusa, VA, USA).

## Figures and Tables

**Figure 1 molecules-26-03189-f001:**
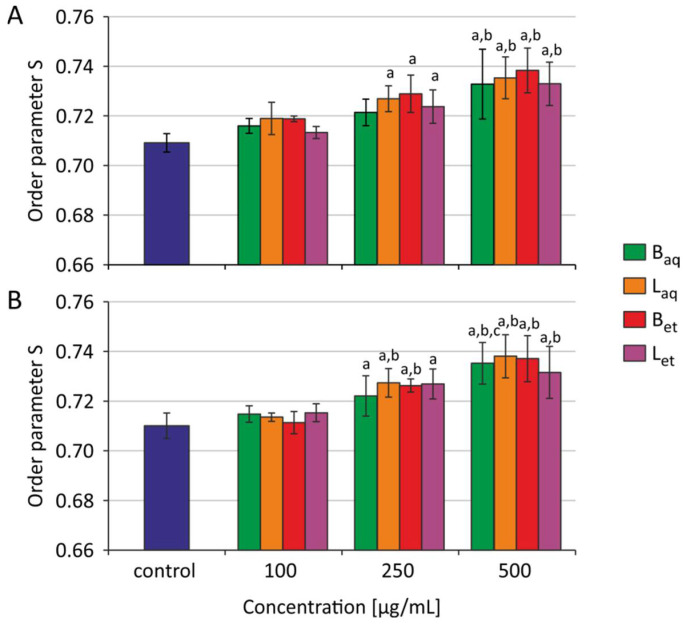
Order parameter S for erythrocytes incubated with extracts from *U. tomentosa* after 2 h (**A**) and 24 h (**B**) (B_aq_–aqueous extract from the bark, L_aq_–aqueous extract from leaves, B_et_–ethanolic extract from the bark, L_et_–ethanolic extract from leaves); ^a^
*p* < 0.05 vs. control, ^b^
*p* < 0.01 vs. control.

**Figure 2 molecules-26-03189-f002:**
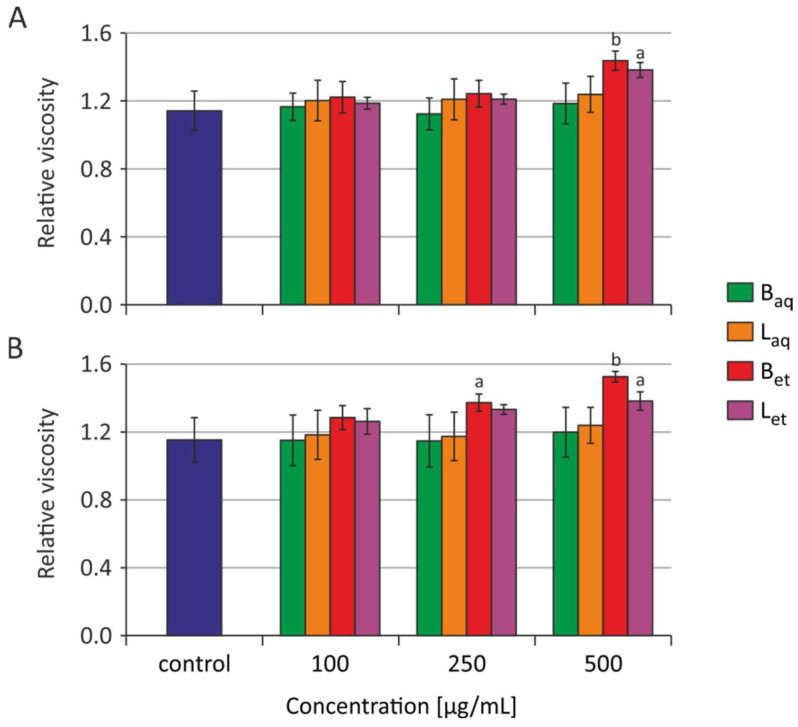
The relative internal viscosity of the erythrocytes incubated with extracts from *U. tomentosa* for 2 h (**A**) and 24 h (**B**) (abbreviations as in Figure 1); ^a^
*p* < 0.05 vs. control, ^b^
*p* < 0.01 vs. control.

**Figure 3 molecules-26-03189-f003:**
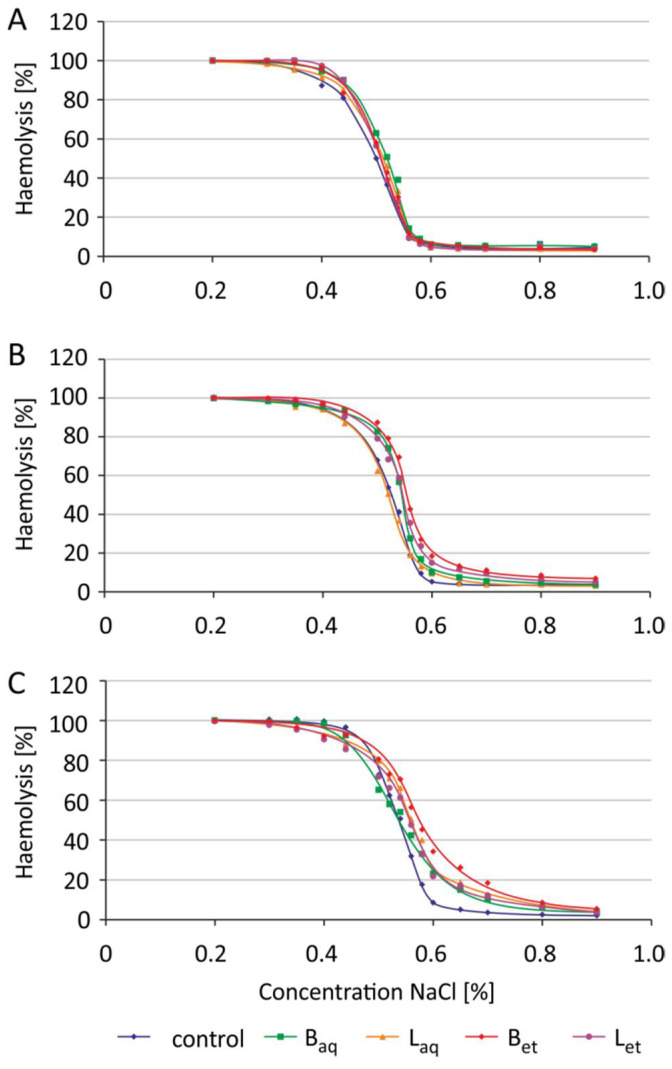
Changes in osmotic resistance of human erythrocytes incubated with *U. tomentosa* extracts at a concentration of 100 μg/mL for 2 h (**A**), 5 h (**B**) and 24 h (**C**) (abbreviations as in Figure 1).

**Figure 4 molecules-26-03189-f004:**
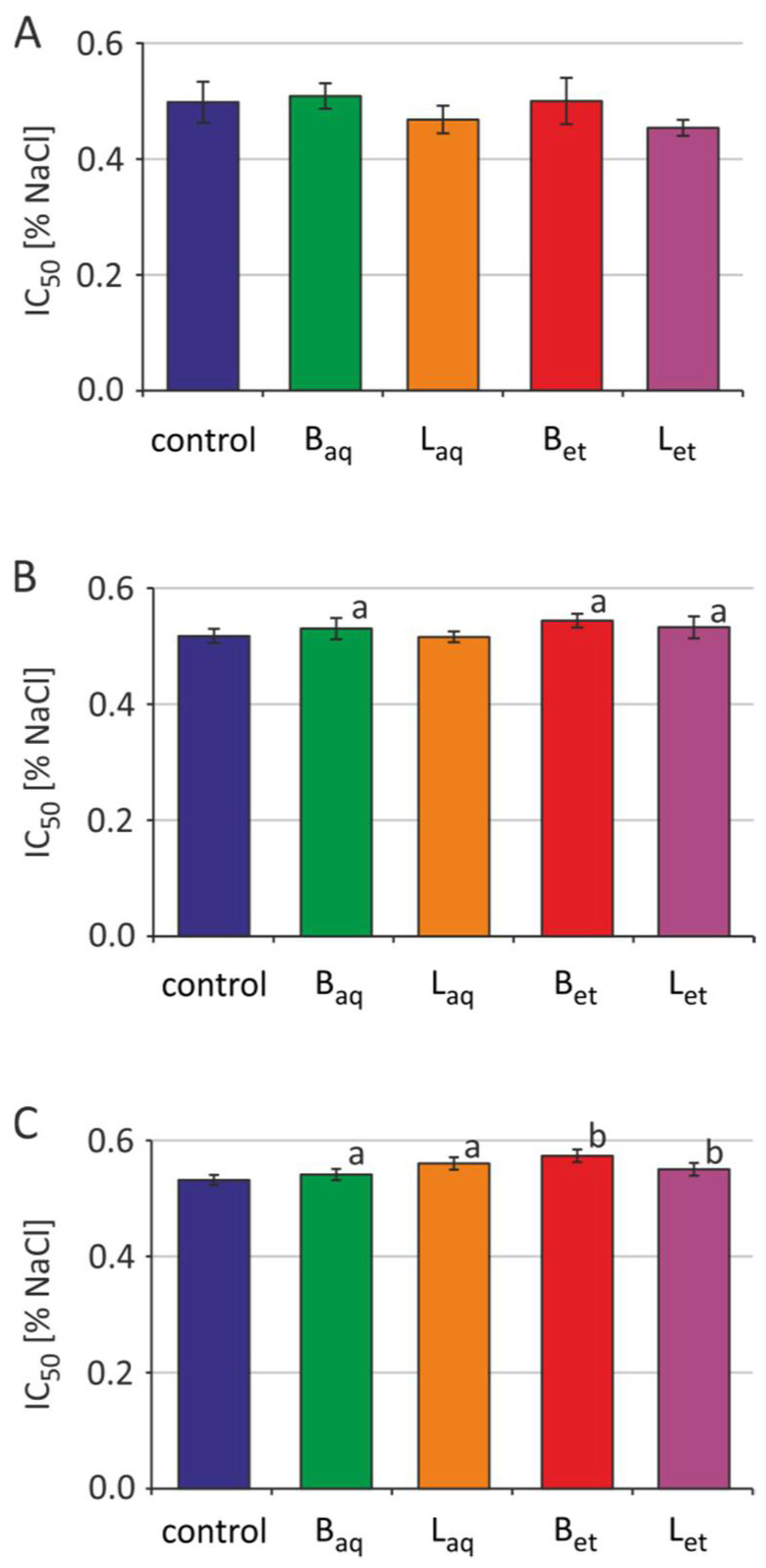
IC_50_ parameter for control RBCs and erythrocytes treated with *U. tomentosa* extracts at a concentration of 100 µg/mL for 2 h (**A**), 5 h (**B**) and 24 h (**C**) (abbreviations as in Figure 1); ^a^
*p* < 0.05; ^b^
*p* < 0.001.

**Figure 5 molecules-26-03189-f005:**
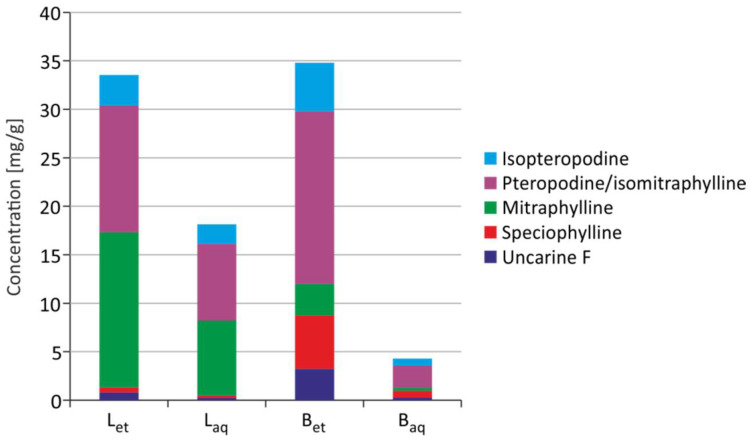
Content of alkaloids in the extracts from *U. tomentosa* (B_et_–ethanolic extract from the bark; B_aq_–aqueous extract from the bark; L_et_–ethanolic extract from leaves; L_aq_–aqueous extract from leaves) [18].

**Figure 6 molecules-26-03189-f006:**
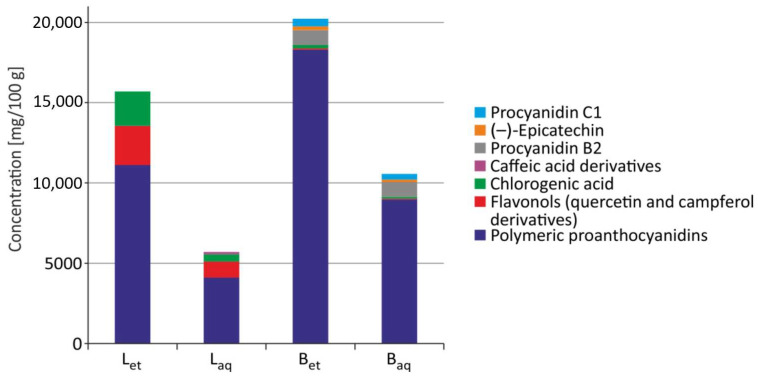
Polyphenols content in the extracts from leaves and bark of *U. tomentosa* (B_et_–ethanolic extract from the bark; B_aq_–aqueous extract from the bark; L_et_–ethanolic extract from leaves; L_aq_–aqueous extract from leaves) [17].

**Table 1 molecules-26-03189-t001:** Correlation time τ_B_ and τ_C_ for erythrocytes incubated with extracts from *U. tomentosa* for 2 h and 24 h (B_aq_–aqueous extract from the bark, L_aq_–aqueous extract from leaves, B_et_–ethanolic extract from the bark, L_et_–ethanolic extract from leaves).

	Concentration	τ_B_ (×10^−10^ s)		τ_C_ (×10^−10^ s)	
	µmol/L	2 h	24 h	2 h	24 h
control		17.636 ± 0.314	17.557 ± 0.307	23.867 ± 0.468	23.757 ± 0.677
	100	17.903 ± 0.308	17.787 ± 0.432	24.151 ± 0.415	23.829 ± 0.602
B_aq_	250	17.476 ± 0.398	17.912 ± 0.690	24.428 ± 0.430	24.123 ± 0.589
	500	18.229 ± 0.445	18.377 ± 0.848	24.552 ± 0.583	24.216 ± 0.669
	100	17.677 ± 0.410	17.611 ± 0.736	23.438 ± 0.527	23.716 ± 0.756
L_aq_	250	17.735 ± 0.537	17.677 ± 0.696	24.204 ± 0.450	23.757 ± 0.659
	500	18.290 ± 0.544	17.873 ± 0.638	24.336 ± 0.391	24.123 ± 0.662
	100	17.480 ± 0.371	17.631 ± 0.691	24.282 ± 0.452	23.910 ± 0.485
B_et_	250	17.972 ± 0.537	17.918 ± 0.718	24.457 ± 0.677	24.171 ± 0.746
	500	18.650 ± 0.468 ^a^	18.803 ± 0.786	25.102 ± 0.521 ^a^	24.620 ± 0.625
	100	17.538 ± 0.430	17.439 ± 0.352	23.673 ± 0.525	23.738 ± 0.507
L_et_	250	17.426 ± 0.309	18.200 ± 0.811	23.962 ± 0.525	23.943 ± 0.642
	500	18.321 ± 0.578	18.605 ± 0.542	24.287 ± 0.533	24.022 ± 0.588

^a^*p* < 0.05 vs. control.

## Data Availability

The data presented in this study are available on request from the corresponding author.

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
