# Peer review of "Changes in Human Erythrocyte Membrane Exposed to Aqueous and Ethanolic Extracts from Uncaria tomentosa"

_molecules, 2021, doi:10.3390/molecules26113189_

Round 1

Reviewer 1 Report

The authors present an interesting study regarding the effects of Uncaria tomentosa on the erythrocyte membrane. However, I have the following comments.

I. Major Comments:
1. In the introduction I suggest including a brief paragraph regarding the effect of oxidative stress on the erythrocyte membrane.

2. Methodology: The cell membrane is made up of phospholipids (fatty acids). I suggest including the analysis of the fatty acid composition. This is an important point, because oxidative stress influences the fatty acid composition of the membrane.

3. In the discussion it is necessary to include a paragraph regarding the effect of oxidative stress and antioxidants on the fatty acid composition of the cell membrane. Especially little eyelids.
Suggested references:

Supplementation with antioxidant-rich extra virgin olive oil prevents hepatic oxidative stress and reduction of desaturation capacity in mice fed a high-fat diet: Effects on fatty acid composition in liver and extrahepatic tissues. Nutrition. 2016; 32: 1254-67.

Hydroxytyrosol prevents reduction in liver activity of Δ-5 and Δ-6 desaturases, oxidative stress, and depletion in long chain polyunsaturated fatty acid content in different tissues of high-fat diet fed mice. Lipids Health Dis. 2017; 16: 64.

II. Minor comments:
1. Improve the writing of the study objective.
2. The resolution of the figures is not good. To get better.
3. Standardize all graphics, for example: fig. 4., there is no space between the columns, while in fig. 6 (space between columns).

Author Response

Major Comments:

In the introduction I suggest including a brief paragraph regarding the effect of oxidative stress on the erythrocyte membrane.

We added this information

Oxidative stress occurs when the level of ROS in a cell overwhelms the cell's antioxidant defence system. The red blood cell is particularly susceptible to oxidative stress due to the high content of polyunsaturated fatty acids present in its membrane and the auto-oxidation of hemoglobin. Human erythrocyte membrane is composed from saturated fatty acids, monounsaturated fatty acids, polyunsaturated fatty acids and long-chain N-3 polyunsaturated fatty acids (Harris et al., 2005). Oxidizing agents change membrane stiffness, alter membrane dipole potential (ψd) and cause the formation of spectrin–hemoglobin complexes (Jewell et al., 2013). To cope with oxidative stress, the erythrocytes are equipped with a specialized cytoskeleton that provides their mechanical stability and flexibility necessary to withstand forces occurring during circulation. Erythrocytes membrane is composed of a lipid bilayer, transmembrane proteins and a filamentous meshwork of proteins (such as spectrin, F-actin, protein 4.1, and actin-binding proteins dematin, adducin, tropomyosin, and tropomodulin) that forms a membrane skeleton along the entire cytoplasmic surface of their membrane (Li et al., 2007). ROS are also capable of inducing eryptosis by changes in membrane phosphatidylserine (PS) translocation (Jarosiewicz et al., 2019). Oxidative stress may also lead to exhaustion of cellular reductive components and structural and functional changes in red blood cell membrane: degradation of phosphatidylserine and sialic acid, which are the main components establishing the negative charge on the outer side of the membrane (Suzuki et al., 2001).

Methodology: The cell membrane is made up of phospholipids (fatty acids). I suggest including the analysis of the fatty acid composition. This is an important point, because oxidative stress influences the fatty acid composition of the membrane.

We added the composition of the fatty acids in to Introduction and discussion

As suggested by the reviewer, we added the information about membrane lipid composition based on the publication of Harris et al.( 2005).  This information is placed in to “Introduction” and at the end of “Discussion” section.

Introduction

Human erythrocyte membrane is composed from saturated fatty acids, monounsaturated fatty acids, polyunsaturated fatty acids and long-chain N-3 polyunsaturated fatty acids (Harris et al., 2005).

Discussion

The limitation of this study is a lack of evaluation of the effect of tested extracts on the fatty acid composition in erythrocyte membrane. Human erythrocyte membrane is com-posed from saturated fatty acids (palmitic, stearic), monounsaturated fatty acids (pal-mitoleic, vacceinic, cis-oleic, trans-oleic), polyunsaturated fatty acids (cis-linoleic, trans-inoleic, linolenic, eicosadenoic, eicosatrienoic, arachosatrienoic) and long-chain N-3 polyunsaturated fatty acids (eicosapentaenoic, docosahexaenoic).

However, in this study, the extracts were applied to erythrocytes obtained from healthy donors. Thus, it can be assumed that the lipid composition of the erythrocyte membrane is typical for normal cells (Harris et al., 2005). Due to the fact that human ma-ture erythrocytes do not have a nucleus, they do not synthesize cellular components, in-cluding fatty acids. For this reason, the lipid composition should not change as opposed to in vivo studies: in squamous cell carcinoma of the skin (Harris et al., 2005) or during supplementation (Rincón-Cervera et al., 2016, Valenzuela et al., 2017). The flip-flop pro-cess, lipid oxidation, change of interactions in the lipid domains and their distribution are possible, but as lipid biosynthesis in erythrocytes does not occur, the qualitative and quantitative composition of the erythrocyte membrane lipids under the influence of ex-tracts tested added in vitro is very unlikely to occur.

This is only in response to the reviewer:

The samples tested were always compared to the control samples from healthy donors and the resulting changes were compared to the changes in control, as in the study of GrÄ™bowski et al. (2013) „After a 48-h incubation in all investigated samples, a fraction of cells with an increased FSC parameter was detected (Fig. 4A). This effect, however, cannot be attributed to the presence of fullerenol, as control cells displayed the same features and no concentration-dependence was observed. Prolonged incubations of erythrocytes led to exhaustion of cellular reductive components and structural and functional changes in the membranes: degradation of phosphatidylserine and sialic acid, which are the main components establishing the negative charge on the outer side of the membrane [51]. All these effects can be a direct result of prolonged incubation at 37 °C and cause significant changes in morphology and size of the erythrocytes” (Grebowski et al., 2013).

  1. Grebowski et al. / Biochimica et Biophysica Acta 1828 (2013) 2007–2014.

In the discussion it is necessary to include a paragraph regarding the effect of oxidative stress and antioxidants on the fatty acid composition of the cell membrane. Especially little eyelids.

Suggested references:

Supplementation with antioxidant-rich extra virgin olive oil prevents hepatic oxidative stress and reduction of desaturation capacity in mice fed a high-fat diet: Effects on fatty acid composition in liver and extrahepatic tissues. Nutrition. 2016; 32: 1254-67.

Hydroxytyrosol prevents reduction in liver activity of Δ-5 and Δ-6 desaturases, oxidative stress, and depletion in long chain polyunsaturated fatty acid content in different tissues of high-fat diet fed mice. Lipids Health Dis. 2017; 16: 64.

We agree with the reviewer that this is an important issue, as it has been shown in the studies suggested for citation. However, in our opinion this is only important in in vivo studies where new blood cells are created with a variable lipid composition depending on supplementation or diseases. Erythrocytes from healthy people (donors) cannot form lipids in vitro because they do not have the nucleus or other organelles. Certainly, the determination of the lipid profile would enrich the work, but in our opinion it  is not of crucial importance in the aspect of the conducted research.

According to the Reviewer suggestion, the following information has been added at the end of “Discussion’”section.

The limitation of this study is a lack of evaluation of the effect of tested extracts on the fatty acid composition in erythrocyte membrane. Human erythrocyte membrane is com-posed from saturated fatty acids (palmitic, stearic), monounsaturated fatty acids (pal-mitoleic, vacceinic, cis-oleic, trans-oleic), polyunsaturated fatty acids (cis-linoleic, trans-inoleic, linolenic, eicosadenoic, eicosatrienoic, arachosatrienoic) and long-chain N-3 polyunsaturated fatty acids (eicosapentaenoic, docosahexaenoic).

However, in this study, the extracts were applied to erythrocytes obtained from healthy donors. Thus, it can be assumed that the lipid composition of the erythrocyte membrane is typical for normal cells (Harris et al., 2005). Due to the fact that human ma-ture erythrocytes do not have a nucleus, they do not synthesize cellular components, in-cluding fatty acids. For this reason, the lipid composition should not change as opposed to in vivo studies: in squamous cell carcinoma of the skin (Harris et al., 2005) or during supplementation (Rincón-Cervera et al., 2016, Valenzuela et al., 2017). The flip-flop pro-cess, lipid oxidation, change of interactions in the lipid domains and their distribution are possible, but as lipid biosynthesis in erythrocytes does not occur, the qualitative and quantitative composition of the erythrocyte membrane lipids under the influence of ex-tracts tested added in vitro is very unlikely to occur.

Minor comments:

Improve the writing of the study objective.

In our opinion, the purpose of the study is described comprehensively

The resolution of the figures is not good. To get better.

High resolution of figures will be send to the journal's editorial office.

Standardize all graphics, for example: fig. 4., there is no space between the columns, while in fig. 6 (space between columns).

For Figure3 and 4 the results from three concentrations of four extracts are presented in one figure. Figure 6 shows the results for one concentration of four extracts. Removing the gaps between the bars may reduce legibility.

References added:

Fernandes, I., Faria, A., Calhau, C., Freitas, V. de, Mateus, N., 2014. Bioavailability of anthocyanins and derivatives. Journal of Functional Foods 7, 54–66. https://doi.org/10.1016/j.jff.2013.05.010.

Harris, R.B., Foote, J.A., Hakim, I.A., Bronson, D.L., Alberts, D.S., 2005. Fatty acid composition of red blood cell membranes and risk of squamous cell carcinoma of the skin. Cancer Epidemiology, Biomarkers & Prevention 14, 906–912. https://doi.org/10.1158/1055-9965.EPI-04-0670.

Jarosiewicz, M., MichaÅ‚owicz, J., Bukowska, B., 2019. In vitro assessment of eryptotic potential of tetrabromobisphenol A and other bromophenolic flame retardants. Chemosphere 215, 404–412. https://doi.org/10.1016/j.chemosphere.2018.09.161.

Jewell, S.A., Petrov, P.G., Winlove, C.P., 2013. The effect of oxidative stress on the membrane dipole potential of human red blood cells. Biochimica et Biophysica Acta 1828, 1250–1258. https://doi.org/10.1016/j.bbamem.2012.12.019.

Li, J., Lykotrafitis, G., Dao, M.; Suresh, S., 2007. Cytoskeletal dynamics of human erythrocyte. Proceeding of the National Academy of Sciences U. S. A. 104, 4937–4942. https://doi.org/10.1073/pnas.0700257104.

Murkovic, M., Adam, U., Pfannhauser, W., 2000. Analysis of anthocyane glycosides in human serum. Fresenius Journal of Analytical Chemistry 366, 379–381. https://doi.org/10.1007/s002160050077.

Rincón-Cervera, M.A., Valenzuela, R., Hernandez-Rodas, M.C., Marambio, M., Espinosa, A., Mayer, S., Romero, N., Barrera M Sc, C., Valenzuela, A., Videla, L.A., 2016. Supplementation with antioxidant-rich extra virgin olive oil prevents hepatic oxidative stress and reduction of desaturation capacity in mice fed a high-fat diet: Effects on fatty acid composition in liver and extrahepatic tissues. Nutrition 32, 1254–1267. https://doi.org/10.1016/j.nut.2016.04.006.

Suzuki, Y., Tateishi, N., Cicha, I., Maeda, N., 2001. Aggregation and sedimentation of mixtures of erythrocytes with different properties. Clinical Hemorheology and Microcirculation 25, 105–117.

Tena, N., Martín, J., Asuero, A.G., 2020. State of the Art of Anthocyanins: Antioxidant Activity, Sources, Bioavailability, and Therapeutic Effect in Human Health. Antioxidants (Basel) 9. https://doi.org/10.3390/antiox9050451.

Valenzuela, R., Echeverria, F., Ortiz, M., Rincón-Cervera, M.Á., Espinosa, A., Hernandez-Rodas, M.C., Illesca, P., Valenzuela, A., Videla, L.A., 2017. Hydroxytyrosol prevents reduction in liver activity of Δ-5 and Δ-6 desaturases, oxidative stress, and depletion in long chain polyunsaturated fatty acid content in different tissues of high-fat diet fed mice. Lipids in Health and  Disease 16, 64. https://doi.org/10.1186/s12944-017-0450-5.

Reviewer 2 Report

I have one substantial question about the methods that I cannot see clearly addressed in the manuscript and is essential to allow effective interpretation of the results. Without this information, it is difficult to determine if the interpretation of the results is accurate so it needs to be included.

I am confused as to how the compounds were resolubilised for addition to the RBCs – line 112 refers to preparation of stock solutions but were these in ethanol, water, buffer or some other solution?  What is the evidence that all of the compounds resolubilse, especially those from the ethanol extractions? What was the control in this case?  This is essential information to be able to interpret the results, due to the effect of ethanol on membranes.

Other queries:

Is the significant difference in table 1 “real” given the number of tests performed and the p value?

It seems unusual that such similar results were obtained with the aqueous and ethanolic extracts.  It seems counterintuitive that molecule readily soluble in water would penetrate to the core of the bilayer – it would be worth discussing the biophysical properties of the identified compounds in this context.

It would be good to see mention of the cytoskeleton as presumably this has a significant role in maintaining cell shape?

The point about the physiological concentration of these compounds is a good one and I think deserves more prominence – is there an estimate for what the physiological concentrations may be?

It would be good to have some commentary on how the proposed alterations in the internal viscosity may be mediated.  The cytoplasm is already very crowded.  Is this due to osmosis and a reduction in water content in the cell, or a specific effect of these compounds?  This could link to the data on osmosis?

Author Response

I have one substantial question about the methods that I cannot see clearly addressed in the manuscript and is essential to allow effective interpretation of the results. Without this information, it is difficult to determine if the interpretation of the results is accurate so it needs to be included.

I am confused as to how the compounds were resolubilised for addition to the RBCs – line 112 refers to preparation of stock solutions but were these in ethanol, water, buffer or some other solution?  What is the evidence that all of the compounds resolubilise, especially those from the ethanol extractions? What was the control in this case?  This is essential information to be able to interpret the results, due to the effect of ethanol on membranes.

The following information was added into the ‘Methods’ section

Finally, all extracts were dissolved in DMSO. The obtained solution was added in a small volume of 1 µL to 1 mL of erythrocytes suspension (DMSO concentration did not influence cell membrane). The erythrocytes supplemented only with DMSO were used as controls.

Other queries:

Is the significant difference in table 1 “real” given the number of tests performed and the p value?

It seems unusual that such similar results were obtained with the aqueous and ethanolic extracts.  It seems counterintuitive that molecule readily soluble in water would penetrate to the core of the bilayer – it would be worth discussing the biophysical properties of the identified compounds in this context.

Although the results obtained for water and ethanolic extracts seem similar, the performed statistical analysis (Anova and Tukey's post hoc test) showed statistical significance only for the values marked in Table 1.

It would be good to see mention of the cytoskeleton as presumably this has a significant role in maintaining cell shape?

According to the Reviewer comment, the role of cytoskeleton was described

Oxidative stress occurs when the level of ROS in a cell overwhelms the cell's antioxidant defence system. The red blood cell is particularly susceptible to oxidative stress due to the high content of polyunsaturated fatty acids present in its membrane and the auto-oxidation of hemoglobin. Human erythrocyte membrane is composed from saturated fatty acids, monounsaturated fatty acids, polyunsaturated fatty acids and long-chain N-3 polyunsaturated fatty acids (Harris et al., 2005). Oxidizing agents change membrane stiffness, alter membrane dipole potential (ψd) and cause the formation of spectrin–hemoglobin complexes (Jewell et al., 2013). To cope with oxidative stress, the erythrocytes are equipped with a specialized cytoskeleton that provides their mechanical stability and flexibility necessary to withstand forces occurring during circulation. Erythrocytes membrane is composed of a lipid bilayer, transmembrane proteins and a filamentous meshwork of proteins (such as spectrin, F-actin, protein 4.1, and actin-binding proteins dematin, adducin, tropomyosin, and tropomodulin) that forms a membrane skeleton along the entire cytoplasmic surface of their membrane (Li et al., 2007). ROS are also capable of inducing eryptosis by changes in membrane phosphatidylserine (PS) translocation (Jarosiewicz et al., 2019). Oxidative stress may also lead to exhaustion of cellular reductive components and structural and functional changes in red blood cell membrane: degradation of phosphatidylserine and sialic acid, which are the main components establishing the negative charge on the outer side of the membrane (Suzuki et al., 2001).

The point about the physiological concentration of these compounds is a good one and I think deserves more prominence – is there an estimate for what the physiological concentrations may be?

As suggested by the reviewer, we added the information in to “Introduction”.

The physiological concentration of plant polyphenols is difficult to  assess. In the literature, usually the information about recommended doses of preparations or  the concentrations of tested compounds (for people 210–500 mg/day or for animals 350–500 mg/kg body mass) is given (Fernandes et al., 2014, Murkovic et al., 2000, Noelia et al., 2020).

Some of the compounds are absorbed without changing their form,  others, especially in the form of glucosides, during absorption are depriving of all or a part of sugar residues. The intestinal flora also participates in the transformation of plant polyphenols.

Most of the absorbed dose is excreted from the body via urine and feces. It is estimated that no more than 10–20% of the ingested dose of quercetin and isoflavones (genistein and daidzein) is detected in plasma after 1–3 hours of administration. Anthocyanins are considered to be less efficiently absorbed – their absorption is estimated at 1–2% of the initial dose.

For reviewer only:

According to PubChem, the alkaloids contained in U. tomentosa belong to the group of indolizines (oxindole alkaloids). Of the compounds mentioned in the work: uncarine F, speciophylline, mitraphylline, pteropodine/isomitraphylline, isopteropodine, only uncarine F is not marked with Acute Toxic symbols. There are no studies on the physiological concentration of these compounds in the literature.

It would be good to have some commentary on how the proposed alterations in the internal viscosity may be mediated.  The cytoplasm is already very crowded.  Is this due to osmosis and a reduction in water content in the cell, or a specific effect of these compounds?  This could link to the data on osmosis?

Alterations in the viscosity of the erythrocyte interior may be associated with a change in the shape of the erythrocytes. This was indicated by changes in the FSC parameter obtained from flow cytometric analysis and photos achieved from a phase-contrast microscopic examination (Bors et al., 2012b).

The formation of echinocytes may result from the placement of polyphenols present in U. tomentosa extracts both in the hydrophilic region and in the hydrophobic region of the outer cell membrane monolayer, as suggested by changes in the ordering parameter S and correlation times τB and τC.

Changes in red blood cell shape may be associated with water loss and the increase of the viscosity inside the cells. Additionally, they may be due to a decrease in the osmotic resistance of the erythrocytes. Placement of altered erythrocytes in an isotonic NaCl solution may lead to faster cell lysis (in comparison to control cells) because of increased effect of water and a limited possibility of deformation of the erythrocyte membrane, and therefore changes of the cell shape.

References added:

Fernandes, I., Faria, A., Calhau, C., Freitas, V. de, Mateus, N., 2014. Bioavailability of anthocyanins and derivatives. Journal of Functional Foods 7, 54–66. https://doi.org/10.1016/j.jff.2013.05.010.

Harris, R.B., Foote, J.A., Hakim, I.A., Bronson, D.L., Alberts, D.S., 2005. Fatty acid composition of red blood cell membranes and risk of squamous cell carcinoma of the skin. Cancer Epidemiology, Biomarkers & Prevention 14, 906–912. https://doi.org/10.1158/1055-9965.EPI-04-0670.

Jarosiewicz, M., MichaÅ‚owicz, J., Bukowska, B., 2019. In vitro assessment of eryptotic potential of tetrabromobisphenol A and other bromophenolic flame retardants. Chemosphere 215, 404–412. https://doi.org/10.1016/j.chemosphere.2018.09.161.

Jewell, S.A., Petrov, P.G., Winlove, C.P., 2013. The effect of oxidative stress on the membrane dipole potential of human red blood cells. Biochimica et Biophysica Acta 1828, 1250–1258. https://doi.org/10.1016/j.bbamem.2012.12.019.

Li, J., Lykotrafitis, G., Dao, M.; Suresh, S., 2007. Cytoskeletal dynamics of human erythrocyte. Proceeding of the National Academy of Sciences U. S. A. 104, 4937–4942. https://doi.org/10.1073/pnas.0700257104.

Murkovic, M., Adam, U., Pfannhauser, W., 2000. Analysis of anthocyane glycosides in human serum. Fresenius Journal of Analytical Chemistry 366, 379–381. https://doi.org/10.1007/s002160050077.

Rincón-Cervera, M.A., Valenzuela, R., Hernandez-Rodas, M.C., Marambio, M., Espinosa, A., Mayer, S., Romero, N., Barrera M Sc, C., Valenzuela, A., Videla, L.A., 2016. Supplementation with antioxidant-rich extra virgin olive oil prevents hepatic oxidative stress and reduction of desaturation capacity in mice fed a high-fat diet: Effects on fatty acid composition in liver and extrahepatic tissues. Nutrition 32, 1254–1267. https://doi.org/10.1016/j.nut.2016.04.006.

Suzuki, Y., Tateishi, N., Cicha, I., Maeda, N., 2001. Aggregation and sedimentation of mixtures of erythrocytes with different properties. Clinical Hemorheology and Microcirculation 25, 105–117.

Tena, N., Martín, J., Asuero, A.G., 2020. State of the Art of Anthocyanins: Antioxidant Activity, Sources, Bioavailability, and Therapeutic Effect in Human Health. Antioxidants (Basel) 9. https://doi.org/10.3390/antiox9050451.

Valenzuela, R., Echeverria, F., Ortiz, M., Rincón-Cervera, M.Á., Espinosa, A., Hernandez-Rodas, M.C., Illesca, P., Valenzuela, A., Videla, L.A., 2017. Hydroxytyrosol prevents reduction in liver activity of Δ-5 and Δ-6 desaturases, oxidative stress, and depletion in long chain polyunsaturated fatty acid content in different tissues of high-fat diet fed mice. Lipids in Health and  Disease 16, 64. https://doi.org/10.1186/s12944-017-0450-5.

Round 2

Reviewer 1 Report

Authors made all changes suggested. Manuscript can be accepted. 

Reviewer 2 Report

I thanks the authors for addressing my comments thoroughly.